# Length of Nutritional Transition Associates Negatively with Postnatal Growth in Very Low Birthweight Infants

**DOI:** 10.3390/nu13113961

**Published:** 2021-11-06

**Authors:** Lotta Immeli, Ulla Sankilampi, Pauliina M. Mäkelä, Markus Leskinen, Reijo Sund, Sture Andersson, Päivi Luukkainen

**Affiliations:** 1Pediatric Research Center, New Children’s Hospital, Helsinki University Hospital, University of Helsinki, 00290 Helsinki, Finland; pauliina.makela@hus.fi (P.M.M.); markus.leskinen@hus.fi (M.L.); sture.andersson@hus.fi (S.A.); paivi.luukkainen@hus.fi (P.L.); 2Department of Pediatrics, Kuopio University Hospital, University of Eastern Finland, 70210 Kuopio, Finland; ulla.sankilampi@kuh.fi; 3Faculty of Health Sciences, Institute of Clinical Medicine, School of Medicine, University of Eastern Finland, 70210 Kuopio, Finland; reijo.sund@uef.fi

**Keywords:** preterm infants, nutrition, very low birthweight infant, transition phase, growth, parenteral nutrition, enteral nutrition

## Abstract

Very low birthweight (VLBW, <1500 g) infants may be predisposed to undernutrition during the nutritional transition phase from parenteral to enteral nutrition. We studied the associations among the length of the transition phase, postnatal macronutrient intake, and growth from birth to term equivalent age in VLBW infants. This retrospective cohort study included 248 VLBW infants born before 32 weeks of gestation and admitted to the Children’s Hospital, Helsinki, Finland during 2005–2013. Daily nutrient intakes were obtained from computerized medication administration records. The length of the transition phase correlated negatively with cumulative energy, protein, fat, and carbohydrate intake at 28 days of age. It also associated negatively with weight and head circumference growth from birth to term equivalent age. For infants with a long transition phase (over 12 d), the estimates (95% CI) for weight and head circumference z-score change from birth to term equivalent age were −0.3 (−0.56, −0.04) and −0.44 (−0.81, −0.07), respectively, in comparison to those with a short transition phase (ad 7 d). For VLBW infants, rapid transition to full enteral feeding might be beneficial. However, if enteral nutrition cannot be advanced, well-planned parenteral nutrition during the transition phase is necessary to promote adequate growth.

## 1. Introduction

A very preterm birth may be regarded as a nutritional emergency, and very low birthweight infants (VLBW, birthweight below 1500 g) need parenteral nutrition (PN) during the first days or weeks of life. Simultaneously, enteral nutrition (EN) with small volumes of milk is started and increased gradually according to tolerance. Nutritional guidelines for both parenteral and enteral intake in preterm infants have been published [1,2,3]. A common goal for nutritional care of VLBW infants is to achieve growth similar to fetal growth, although this has recently been debated [3]. Still, extrauterine growth retardation is common in VLBW infants and suboptimal nutrition and growth have been associated with adverse short- and long-term outcomes [4,5,6,7].

VLBW infants undergo a transition phase from parenteral to enteral nutrition. However, guidelines or consensus on how to optimize nutritional care during the transition phase is missing, and practices vary across centers. Data on how this transition phase in VLBW infants affects nutritional intake and postnatal growth remain scarce. It is not known, for example, how the length of the transition phase affects macronutrient intake and postnatal growth.

We aimed to study how the length of the transition phase from parenteral to enteral nutrition associates with macronutrient intake and postnatal growth in VLBW infants. We hypothesized that, in VLBW infants, a prolonged transition phase from parenteral to enteral nutrition is associated with deficits in macronutrient intake and impaired growth. The primary endpoint was growth from birth to the term equivalent age and the secondary endpoint was growth from birth to the age of 28 days.

## 2. Materials and Methods

### 2.1. The Study Design and Population

This retrospective cohort study is part of the “Big data-Tiny infants” research project.

In this study, we included all VLBW infants with a gestational age below 32 weeks admitted to the Neonatal Intensive Care Unit of the Helsinki University Hospital within the first 24 h of life during 2005–2013 and hospitalized in the unit for at least 28 days (*n* = 348). Exclusion criteria included necrotizing enterocolitis, gastrointestinal surgery, major congenital malformations, and chromosomal anomalies (*n* = 65). Infants who did not achieve full enteral feeds by 28 days of age (*n* = 7), or who had missing growth data at term equivalent age (*n* = 28) were also excluded. Thus, the final study population included 248 infants. The Register authority and the Ethics Committee of the Helsinki University Hospital approved the study protocol. As all the data obtained were pseudonymised, informed consent was waived.

### 2.2. Clinical Characteristics

Data on the background characteristics and postnatal morbidities, except for sepsis, were retrieved from the Finnish Medical Birth Register, which is managed by the Finnish Institute for Health and Welfare. A detailed description of the Medical Birth Register data used has been described previously [8,9]. Gestational age was determined from the first day of the last menstrual period and confirmed by ultrasonography in 84% of cases. Being small for gestational age was defined as a birthweight z-score below −2 standard deviations on the Finnish birth size reference [10]; respiratory distress syndrome as presentation of clinical symptoms and typical X-ray findings; bronchopulmonary dysplasia as the need for supplementary oxygen at 36 weeks of postmenstrual age; and necrotizing enterocolitis as Bell stage II or III [11]. Patent ductus arteriosus was diagnosed with echocardiography and intraventricular hemorrhage with ultrasonography using Papile’s classification [12]. A diagnosis of sepsis was based on the laboratory results in the electronic medical records and defined as a positive blood culture together with a C-reactive protein value of more than 10 mg/L on the day the positive blood culture sample was drawn or on the two succeeding days [13].

### 2.3. Nutrition Data

The actual daily enteral and parenteral intakes for each infant were calculated using the computerized medication administration records in the electronic patient information system used in the unit during the study period (Centricity Critical Care Clinisoft, GE Healthcare) together with the manufacturer´s product information. During the study period the composition of human milk was not analysed in our unit, therefore the content for human milk (mother’s own milk or donor milk) was based on the values reported in the literature: 67.7 kcal/100 mL, 1.4 g/100 mL, 3.2 g/100 mL, 8.4 g/100 mL for energy, protein, fat, and carbohydrates, respectively [14]. The daily intake was adjusted by weight using the birthweight for the first seven days and thereafter with the recorded weight of the respective day. The daily intakes for the first 28 days of life were analyzed, and each day comprised full 24-h periods, except for 59 infants whose first day comprised 17.9–23 h.

Before analysis, the nutritional data were screened for possible defects and outliers. There were six days (0.09%) with missing nutritional data and 14 outliers (0.2%) with a daily fluid, energy, or protein intake 2–3 times higher or 8–10 times lower than the intake of the surrounding days of the respective infant. These defects and outliers were interpolated using the surrounding measurements of the respective infant. 

### 2.4. Growth Data

The growth parameters for the first 28 days were retrieved from the electronic medical records. According to our unit’s policy, the weight was recorded at least once a day with the incubator’s in-bed scale, and length and head circumference once a week with a length measuring board and a measuring tape. If more than one growth parameter for a single day was available, the first recorded parameter of the respective day was used. 

The growth parameters at term equivalent age were retrieved from the Finnish Medical Birth Register, and the postmenstrual age at the time of measurements varied between 35+^2/7^ and 43+^6/7^ weeks (median 40+^1/7^, interquartile range 38+^3/7^–41+^6/7^). All the growth parameters were converted to age- and gender-specific z-scores adopting the Finnish growth charts [10]. Two boys were excluded from the length analysis as their length (cm) parameters at term equivalent age were regarded as incorrect. 

### 2.5. Nutrition Practices

During the study period, our unit followed local PN and EN guidelines which were in line with the European PN guideline published in 2005 [15] and the European EN guideline published in 2010 [3]. According to the local guidelines, the PN target for daily energy, protein, and lipid intake was 100–120 kcal/kg, 3.5 g/kg, and 2–3 g/kg, respectively. In EN, the target for daily energy and protein intake was 120–140 kcal/kg and 3.0–3.8 g/kg, respectively, with the highest protein target of 3.6–3.8 g/kg for extremely low birthweight (ELBW, birthweight below 1000 g) infants. After 2010, the daily enteral protein intake target for ELBW infants was updated to 4–4.5 g/kg. 

According to our unit´s policy, parenteral nutrition was started immediately after birth as soon as intravenous access was available, and it was generally discontinued when enteral feeds exceeded 120 mL/kg/d. During the study period, the implementation of the PN varied between individual solutions, standard 2-in-1, and triple-chamber PN solutions [8]. Enteral nutrition was started during the first day of life with minimal amounts of the mother’s own milk or donor human milk, and the amount was gradually increased by 10–20 mL/kg/d, according to tolerance. The milk was fortified when the intake was at least 100 mL/kg/d (Nutriprem BMF or Nutrilon BMF, Nutricia Medical Oy, Turku, Finland). For each infant, the nutrition was prescribed individually with a computerized order entry system, and the tolerated total fluid intake and the local guidelines guided individual variation.

For this study, the nutritional management was divided into three different phases according to the enteral milk intake. The PN phase was defined as a period when nutrition was provided mainly parenterally with enteral feeds of <20 mL/kg/d; the transition phase as a period when enteral feed volumes were advancing between 20 and 120 mL/kg/d; and the EN phase as a period when the milk intake was at least 120 mL/kg/d.

### 2.6. Statistical Analysis

Descriptive data are presented as number (%), mean and standard deviation (SD), or median and interquartile range (IQR) as appropriate. The differences between sexes or subgroups were compared using the Student’s *t*-test, analysis of variance, the Mann–Whitney U-test, the Kruskal–Wallis test, the chi-squared test, or Fisher’s exact test where appropriate. For pairwise comparisons, Tukey’s or Dunn’s test with Bonferroni correction was applied. The significance level was set at 0.05. Spearman correlation was used to study the associations between nutrient intake and transition phase length.

Growth (weight, length, and head circumference) was assessed at the age of one, two, three, and four weeks and at term equivalent age, and all the parameters were converted to age- and sex-specific z-scores. The growth z-score differences between the postnatal time points and birth were calculated. The primary outcomes were the weight, length, and head circumference z-score differences between term equivalent age and birth, and the secondary outcomes were those between 28 days of age and birth.

First, linear mixed models were constructed to examine associations between transition phase length and growth outcomes at all different time points in girls and boys separately. In the model, infants were divided into three groups according to transition phase length quartiles, where the middle quartiles were combined into one group and the first quartile was set as reference level. The role of the transition phase length was explored after adjustments for subject (random effect), and gestational age at birth, weight, length or head circumference z-score at birth, and head circumference model also for severe (grade III or IV) intraventricular hemorrhage (fixed effects). Next, to investigate the association of transition phase length with the primary or the secondary outcome, a linear model was constructed. This model was adjusted for sex, weight, length, or head circumference z-score at birth, gestational age at birth, and head circumference model also for severe (grade III or IV) intraventricular hemorrhage. In addition, all the primary outcome models were adjusted for postmenstrual age at the time of term equivalent age assessments. The statistical analyses were performed using the R version 3.6.2 (R Foundation for Statistical Computing, Vienna, Austria, 2005). 

## 3. Results

### 3.1. Clinical Characteristics

The final analysis included 248 infants (131 boys, 53%) with a total of 10,230 growth assessments, and 6944 daily intakes based on 637,130 medication administration records. The PN phase accounted for 18%, the transition phase for 37%, and the EN phase for 45% of the analyzed daily nutritional intakes. 

The clinical characteristics of the infants are shown in Table 1. The majority (79%) of the study subjects were born extremely preterm (gestational age below 28 weeks) with a birthweight below 1000 g. The girls had a lower mean birthweight and a lower head circumference z-score at birth compared with the boys. There were proportionally more ELBW girls than boys (*p* < 0.05). Otherwise, the clinical characteristics were similar between the sexes, except for bronchopulmonary dysplasia and sepsis whose prevalence was higher among boys. This difference was not, however, statistically significant. 

The median length of the transition phase was 10 days, and girls transitioned to EN one day faster compared with boys (*p* < 0.05). For further analyses, the infants were divided into three groups according to the transition phase length quartiles: short transition (up to 7 days) corresponding to the first quartile, intermediate transition (8 to 12 days) corresponding to the second and third quartile, and long transition (over 12 days) corresponding to the fourth quartile.

Energy and protein intake in the parenteral, transition, and enteral phase are shown in Appendix A. Since the enteral feeding guidelines for ELBW infants were updated in 2010, the data are presented separately for infants born in 2005–2009 and 2010–2013. Infants born in these two eras had similar protein intake during the transition phase.

### 3.2. Cumulative Nutrient Intake and Longitudinal Growth According to the Transition Phase Length

Figure 1 shows the correlations between the transition phase length and the cumulative nutrient intakes at 28 days of age according to sex. The length of the transition phase correlated negatively with the cumulative energy, protein, fat, and carbohydrate intake at 28 days of age in both girls and boys (correlation coefficients between −0.58 and −0.18, *p* < 0.05). 

Table 2 shows the cumulative nutrient intakes and clinical characteristics in girls and boys in the three transition length-based groups. Girls and boys with a short transition phase had the highest cumulative energy, protein, and lipid intake at 28 days of life. Girls with the intermediate transition phase had the lowest gestational age and birthweight, and the longest duration of invasive mechanical ventilation. Boys in the three transition length-based groups had similar gestational age, birthweight, and duration of invasive mechanical ventilation. However, the prevalence of ELBW and being small for gestational age was highest among boys with a short transition, *p* = 0.44 and *p* = 0.2, respectively. The prevalence of postnatal morbidities (respiratory distress syndrome, bronchopulmonary dysplasia, patent ductus arteriosus, sepsis, and severe intraventricular hemorrhage) in girls and boys in the three transition length-based groups is shown in Appendix A. Boys with a short transition had the highest incidence of respiratory distress syndrome (*p* = 0.9), bronchopulmonary dysplasia (*p* = 0.42) and severe intraventricular hemorrhage (*p* = 0.08). In contrast, girls with a short transition had the lowest incidence of respiratory distress syndrome (*p* = 0.9) and bronchopulmonary dysplasia (*p* = 0.05). 

Figure 2 shows the weight, length, and head circumference z-score change from birth to the age of one, two, three, and four weeks, and to term equivalent age in girls and boys in the three transition length-based groups. Girls with a short transition phase (3–7 days), had less faltering in head circumference growth at the age of four weeks compared with those with an intermediate (8–12 days) or a long transition phase (13–24 days) (*p* < 0.05). At the age of four weeks, girls with a short transition phase also tended to have less faltering in weight and length growth compared with those with a long transition phase. This was not, however, statistically significant.

Boys with a short transition phase (4–7 days) had less weight faltering and better head circumference growth at term equivalent age compared with those with an intermediate (8–12 days) or a long transition phase (13–22 days) (*p* < 0.05). The mean head circumference z-score at term equivalent age was 0.12, −0.40, and −0.58 for boys with a short, intermediate, and long transition phase, respectively. Boys with a short transition phase also had less faltering in head circumference growth at the age of three and four weeks compared with those with a longer transition phase. This difference was, however, statistically significant only if compared with the group of boys with an intermediate transition phase length (*p* < 0.05).

### 3.3. Primary and Secondary Outcome

Table 3 shows the results for the primary outcome, the z-score changes from birth to term equivalent age. A long transition phase associated negatively with both weight and head circumference growth from birth to term equivalent age; compared with infants with short transition the estimate of a long transition to weight and head circumference z-score change was −0.3 and −0.44, respectively. For head circumference intermediate transition also had a negative association. Other factors associating negatively with weight, length, or head circumference z-score change from birth to term equivalent age, were higher weight, length, or head circumference z-score at birth, and younger age at the term equivalent age growth assessment. Severe (grade III or IV) intraventricular hemorrhage associated with a larger head circumference at term equivalent age.

Table 3 also shows the results for the secondary outcome, z-score changes from birth to the age of 28 days. A long transition phase associated negatively with weight z-score change from birth to 28 days of age. However, for the head circumference z-score change, only the intermediate transition had a significant negative association. Similarly to the primary outcome, a higher weight, length, or head circumference z-score at birth associated negatively with the growth outcome at 28 days of age, and severe intraventricular hemorrhage with a bigger head circumference at 28 days of age. Male sex associated positively with weight, length, and head circumference z-score change from birth to 28 days of age. 

## 4. Discussion

This study showed that in VLBW infants, the length of the transition phase from parenteral to enteral nutrition was negatively associated with cumulative energy, protein, lipid, and carbohydrate intake at 28 days of age. The length of the transition phase was also associated negatively with postnatal weight and head circumference growth from birth to term equivalent age. 

Previously, substantial macronutrient and energy deficits during the transition phase were reported in a study of 59 VLBW infants [16], and a declining protein intake in a study of 115 VLBW infants [17]. Our results confirm the previous observations and show a linear correlation between the length of the transition phase and nutritional deficiencies. 

To our knowledge, there are only four studies on the role of transition phase in nutrition and postnatal growth in VLBW infants. One study reported a decreased protein intake and compromised weight gain during the transition phase in 156 preterm infants with a birthweight below 2000 g. The compromised weight gain during the transition phase was predictive of growth failure (weight <10th percentile) at discharge [18]. Later, the same group reported improved growth rates during transition phase when nutrition was optimized with the use of a concentrated PN solution [19]. Another study of 106 VLBW infants demonstrated that adequate growth velocity (≥15 g/kg/d) during the transition phase associated with a higher fat-free mass deposition at term equivalent age. However, there was no differences in the anthropometric measurements at term equivalent age when infants with an adequate growth velocity were compared with those with an inadequate growth velocity (<15 g/kg/d) [20]. We demonstrate, for the first time in a large cohort of VLBW infants including mostly ELBW infants, a significant negative association between the length of the transition phase and postnatal weight and head circumference growth. 

For a clinician taking care of VLBW infants, it is important to recognize the possible risk of inadequate nutrition and poor growth associated with the nutritional transition phase when weaning from parenteral nutrition is taking place and enteral feeding is still advancing. While there are international guidelines for parenteral and enteral nutrition for VLBW infants, a guideline for optimal nutrition during the transition phase is lacking. During the study period, our unit used a computerized prescriber order entry program which helped clinicians to order PN also during the transition phase. However, the PN or EN guidelines were not included in the program and the clinician had to enter the nutritional targets manually into the system for each infant. Computer-assisted prescribing software based on the nutritional guidelines could improve the quality of nutritional care in particular during the challenging transition phase [21]. The use of a standard PN solution with optimal composition for the transition phase could also be beneficial [8,19,22,23,24]. It is also important to support the enteral nutrition, prefer the mother’s own milk if available, fortify the human milk early enough, and when possible, advance the enteral volumes towards full feeds actively to avoid a long transition phase [25]. 

We also reported sex-specific differences in the association with transition phase length and postnatal growth. In boys, the longer transition seemed to associate more negatively with postnatal weight and head circumference growth, while in girls the growth was more similar between girls with different transition phase lengths. This is in line with previously reported studies that boys seem to benefit more from dietary interventions [26,27]. However, in a study of 95 ELBW infants, a significant correlation of total calorie intake with a change in weight percentiles during the transition phase was reported in girls only [28]. Contrary to our work, this study only included infants appropriate for gestational age. Further studies are needed to understand the sex-specific differences in the nutrition and growth of preterm infants.

The postnatal period is a time of rapid brain growth, especially of the cerebellum [29]. Thus, the association of a long transition phase with lower macronutrient intake during the first four weeks of life and poorer head growth from birth to term equivalent age is noteworthy. It has been reported previously that in preterm infants, there is a positive association between nutrient intake during the first four postnatal weeks, brain volumes at term equivalent age, and neurodevelopmental outcome at the age of two years [6]. In preterm infants born at less than 30 weeks of gestation, higher energy and lipid intake during the first two postnatal weeks have also been associated with a lower incidence of brain lesions and dysmaturation in magnetic resonance imaging at term equivalent age [30]. However, this study concentrated on the longitudinal growth of VLBW infants until the term equivalent age, and unfortunately the neuroimage findings or later developmental outcomes were not studied. 

The strength of the present study is the cohort size of 248 VLBW infants and the use of comprehensive electronic nutritional and growth data, together with electronic data processing [21,31]. A total of over 600,000 medication administration records and 10,000 growth assessments were analyzed. However, a limiting factor is the eight-year study period, as there may be variation in the neonatal intensive care practices. For instance, during the study period, the implementation of PN varied between individual PN solutions, two-in-one PN solutions, and commercial triple-chamber PN solutions. The local PN guidelines remained unchanged, however, the EN guideline was updated in 2010 and the target protein intake for ELBW infants increased. Yet this did not affect these infants protein intake during the transition phase. A notable limiting factor is that the composition of the human milk used in the analyses was based on values reported in the literature and not on actual analyses. 

Human growth is a highly complex process and various factors, such as general morbidity, chronic low-level inflammation, microbiome, and genetic and epigenetic mechanisms, may affect the postnatal growth of a premature infant [32,33,34]. The infants in this study were mainly vulnerable ELBW infants, of whom more than 20% were growth restricted at birth. Thus, there may have been several underlying confounding factors affecting their growth. Due to the retrospective study design, it was not possible to account for all possible confounding factors in the analyses. However, all the analyses were adjusted for gestational age and the growth z-scores at birth and adjusting for single morbidities did not affect the model outcome. The transition phase is a challenging period in the nutritional care of a VLBW infant. In the future, more research is needed to understand the underlying mechanisms affecting postnatal growth to be able to provide optimal postnatal care, including nutritional care, for VLBW infants.

## 5. Conclusions

We found that the length of the transition phase associated negatively with postnatal macronutrient intake and growth in VLBW infants. For these infants, a rapid transition to full enteral feeding might be beneficial. Therefore, if enteral nutrition cannot be advanced, well-planned parenteral nutrition during the transition phase is necessary.

## Figures and Tables

**Figure 1 nutrients-13-03961-f001:**
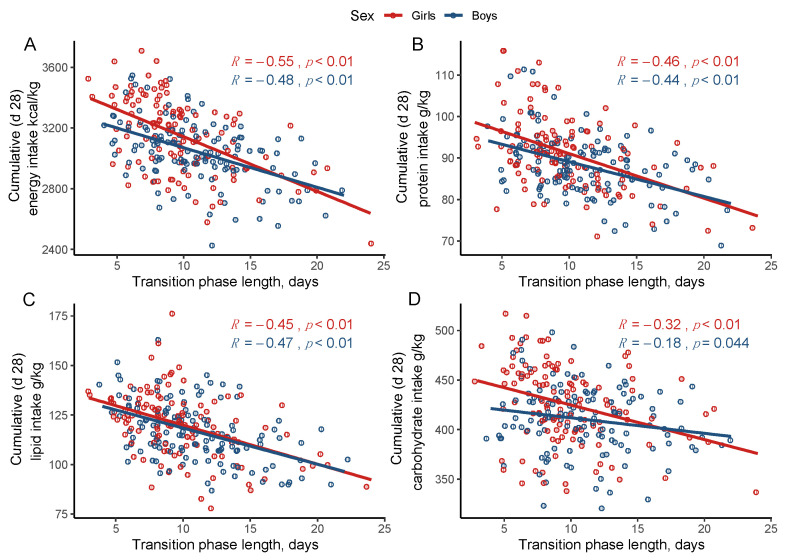
Correlations between transition phase length and cumulative energy (**A**), protein (**B**), lipid (**C**), and carbohydrate (**D**) intake at 28 days of age in girls and boys. Linear regression line is shown together with the 95% confidence interval (shaded). R corresponding to Pearson’s correlation coefficients.

**Figure 2 nutrients-13-03961-f002:**
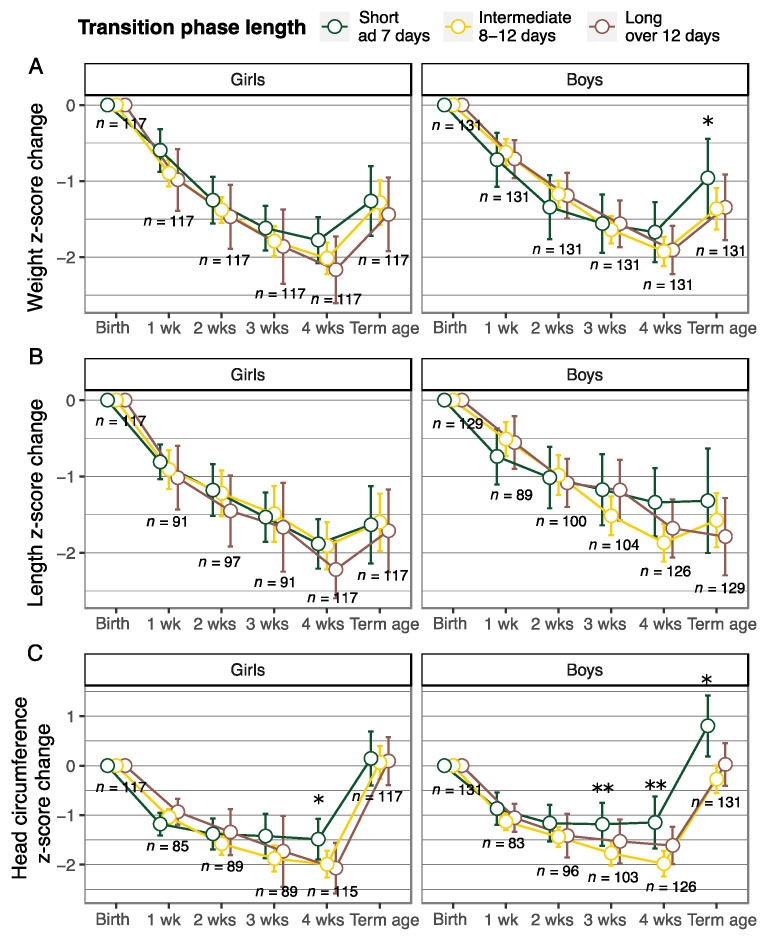
Weight (**A**), length (**B**), and head circumference (**C**) z-score change (mean and 95% confidence interval) from birth to the age of one, two, three, and four weeks, and term equivalent age according to the transition phase length quartiles (intermediate transition corresponding to the second and third quartile). Linear mixed model, adjusted for gestational age at birth and weight, length or head circumference z-score at birth, and head circumference model also for severe (grade III or IV) intraventricular hemorrhage. Short transition length up to 7 days was set as reference level. * *p* < 0.05 for short vs. intermediate and long transition ** *p* < 0.05 for short vs. intermediate transition.

**Table 1 nutrients-13-03961-t001:** Clinical characteristics of the study population.

	Total	Girls	Boys	*p*-Value
Number of infants, *n* (%)	248	117 (47.2)	131 (52.8)	0.37 ^a^
Gestational age, weeks, mean (SD)	26.5 (1.6)	26.4 (1.5)	26.6 (1.8)	0.60 ^b^
Birthweight, g, mean (SD)	845 (199)	798 (170)	887 (213)	0.0003 ^b,^*
Birthweight < 1000 g, *n* (%)	198 (79.8)	104 (88.9)	94 (71.8)	0.001 ^a,^*
Small for gestational age, *n* (%)	51 (20.6)	28 (23.9)	23 (17.6)	0.28 ^a^
Birthweight, z-score, median (IQR)	−0.2 (−1.8–0.9)	−0.4 (−2.0–0.7)	0.0 (−1.8–0.9)	0.12 ^c^
Length at birth, z-score, median (IQR)	−0.6 (−1.9–0.6)	−0.6 (−1.7–0.4)	−0.4 (−2–0.7)	0.6 ^c^
HC at birth, z-score, mean (SD)	−0.5 (1.3)	−0.7 (1.3)	−0.4 (1.3)	0.04 ^b^
Multiple gestation (twins), *n* (%)	53 (21.4)	24 (20.5)	29 (22.1)	0.88 ^a^
Duration of invasive mechanical ventilation, days, median (IQR)	15 (5–33)	13 (5–30)	17 (5–34.5)	0.18 ^c^
Respiratory distress syndrome, *n* (%)	168 (67.7)	79 (67.5)	89 (67.9)	1 ^a^
Bronchopulmonary dysplasia, *n* (%)	127 (51.2)	52 (44.4)	75 (57.3)	0.059 ^a^
Patent ductus arteriosus, *n* (%)	165 (66.5)	80 (68.4)	85 (64.9)	0.66 ^a^
Sepsis, *n* (%)	59 (23.8)	23 (19.7)	36 (27.5)	0.2 ^a^
Severe IVH (grade III or IV), *n* (%)	27 (10.9)	13 (11.1)	14 (10.7)	1 ^a^
Discharged before 42 weeks of PMA, *n* (%)	184 (74.2)	88 (75.2)	96 (73.3)	0.62 ^a^
Parenteral phase length, d, median (IQR)	5 (3–7)	5 (2–7)	5 (3–7.5)	0.7 ^c^
Transition phase length, d, median (IQR)	10 (8–12.3)	9 (7–11)	10 (8–13)	0.007 ^c,^*
Enteral phase length, d, median (IQR)	13 (9–16)	14 (10–17)	12 (8–15)	0.02 ^c,^*

HC = head circumference, IVH = intraventricular hemorrhage, PMA = postmenstrual age. Small for gestational age is defined as a birthweight z-score below −2 standard deviations. a = chi-squared test, b = *t*-test, c = Mann–Whitney U-test, * *p* < 0.05.

**Table 2 nutrients-13-03961-t002:** Cumulative nutrient intake at 28 days of age and clinical characteristics of girls and boys according to the transition phase length quartiles.

	Girls (*n* = 117)				Boys (*n* = 131)			
	Short Transition ad 7 Days (*n* = 31)	Intermediate Transition 8–12 Days (*n* = 64)	Long Transition over 12 Days (*n* = 22)	*p*-Value	Short Transition ad 7 Days (*n* = 23)	Intermediate Transition 8–12 Days (*n* = 68)	Long Transition over 12 Days (*n* = 40)	*p*-Value
Cumulative nutrient intake at 28 days of age, mean (SD)								
energy kcal/kg	3299 (223)	3142 (230)	2971 (217)	<0.05 ^b,^*	3214 (188)	3058 (187)	2930 (181)	<0.05 ^b,^*
protein g/kg	95.26 (9.9)	91.4 (7.4)	85.5 (7.9)	<0.05 ^b,^**	94.7 (8.1)	88.5 (5.6)	84.5 (6.9)	<0.05 ^b,^*
lipid g/kg	126.6 (9.5)	121.5 (17.9)	108.4 (14.2)	<0.05 ^b,^**	126.1 (13.0)	118.6 (14.0)	108.3 (13.3)	<0.05 ^b,^**
Carbohydrate g/kg	444.7 (45.6)	420.8 (36.1)	413.4 (39.7)	<0.05 ^b,^***	425.1 (33.9)	409.0 (34.6)	404.5 (30.6)	0.06 ^b^
Clinical characteristics								
Gestational age, wk, mean (SD)	26.4 (1.5)	25.6 (1.4)	26.6 (1.6)	<0.05 ^b,^^	26.0 (1.8)	26.0 (1.6)	26.4 (2.1)	0.5 ^b^
Birthweight, g, mean (SD)	805 (143)	764 (161)	886 (203)	<0.05 ^b,^^	865 (263)	897 (201)	883 (205)	0.8 ^b^
Birthweight < 1000 g, *n* (%)	28 (90.3)	59 (92.2)	17 (77.3)	0.15 ^a^	19 (82.6)	47 (69.1)	28 (70.0)	0.44 ^a^
Small for GA, *n* (%)	8 (25.8)	14 (21.9)	6 (27.3)	0.8 ^a^	7 (30.4)	9 (13.2)	7 (17.5)	0.2 ^a^
Duration of invasive mechanical ventilation, days median (IQR)	7 (3.5–16)	22 (8–35.3)	10 (3–27.3)	<0.05 ^c,^****	20 (4–36)	17.5 (6.5–34.3)	17 (5.8–32.8)	0.98 ^c^

a = chi-squared test, b = analysis of variance, c = Kruskal–Wallis test. The pairwise analysis with Tukey’s or Dunn’s test with Bonferroni correction. * *p* < 0.05 for all pairwise comparisons. ** *p* < 0.05 for short and intermediate versus long transition. *** *p* < 0.05 for short vs. long transition. **** *p* < 0.05 for short vs. intermediate transition. ^ *p* < 0.05 for intermediate vs. long transition.

**Table 3 nutrients-13-03961-t003:** Associations between transition phase length and postnatal growth (linear model). Short transition length up to 7 days was set as reference level.

	Change in z-Score from Birth to Term Equivalent Age ^			Change in z-Score from Birth to 28 Days of Age		
	Weight	Length	Head Circumference	Weight	Length	Head Circumference
	Coefficient (95% CI)			Coefficient (95% CI)		
Intermediate transition (8–12 d)	−0.13 (−0.35, 0.09)	0.03 (−0.32, 0.39)	−0.43 (−0.74, −0.11) *	−0.12 (−0.27, 0.03)	−0.18 (−0.45, 0.093)	−0.49 (−0.79, −0.20) *
Long transition (over 12 d)	−0.3 (−0.56, −0.04) *	−0.25 (−0.67, 0.16)	−0.44 (−0.81, −0.07) *	−0.22 (−0.4, −0.044) *	−0.23 (−0.54, 0.086)	−0.34 (−0.69, 0.011)
Gestational age (week)	0.03 (−0.03, 0.09)	0.19 (0.10, 0.28) *	0.082 (−0.0009, 0.17)	−0.06 (−0.10, −0.02) *	0.004 (−0.064, 0.073)	0.10 (0.022, 0.18) *
Weight/length/head circumference at birth (z-score)	−0.53 (−0.59, −0.47) *	−0.45 (−0.53, −0.37) *	−0.58 (−0.68, −0.47) *	−0.47 (−0.51, −0.43) *	−0.35 (−0.41, −0.29) *	−0.38 (−0.48, −0.28) *
Sex (boys)	0.11 (−0.07, 0.29)	−0.09 (−0.37, 0.19)	0.093 (−0.16, 0.34)	0.26 (0.14, 0.38) *	0.26 (0.04, 0.47) *	0.30 (0.065, 0.54) *
PMA (week) at the time of the measurement ^	0.16 (0.11, 0.21) *	0.16 (0.087, 0.24) *	0.14 (0.077, 0.21) *	–	–	–
Severe IVH	–	–	0.55 (0.14, 0.95) *	–	–	0.80 (0.41, 1.18) *
R^2^	0.67	0.49	0.45	0.70	0.38	0.32

The coefficients are changes in the z-score. Infants are divided into three groups according to the transition phase length quartiles: Short transition (reference) corresponding to the first quartile, intermediate transition to the second and third quartile, and long transition to the fourth quartile. * *p*-value < 0.05, ^ The growth parameters at term equivalent age were obtained from the Finnish Medical Birth Register and the postmenstrual age (PMA) at the time of the measurement varied between 35+^2/7^ and 43+^6/7^ weeks (median 40+^1/7^). Severe IVH corresponds to intraventricular hemorrhage grade III or IV.

## Data Availability

Data available on request due to privacy/ethical restrictions.

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
