# Peer review of "Length of Nutritional Transition Associates Negatively with Postnatal Growth in Very Low Birthweight Infants"

_nutrients, 2021, doi:10.3390/nu13113961_

Round 1
Reviewer 1 Report
This is a very well-written and interesting manuscript. Thank you for the opportunity to review. I only had one comment in the introduction: It would be helpful to note here what is the usual practice or recommendation on the timing of the transition from parenteral to enteral feeding or the differences for different circumstances so that the reader can have an understanding of what is currently done.
Author Response
We thank the reviewer for this comment. A universal recommendation for nutritional transition is missing and the practices vary across centers. To point this out, we have now revised the introduction to (lines 48–50): “However, guidelines or consensus on how to optimize nutritional care during the transition phase is missing, and the practices vary across centers.”
Reviewer 2 Report
The authors conducted a retrospective study to explore the association between length of the transition phase (TF) from enteral to parenteral nutrition, macronutrient intake and postnatal growth in VLBW infants. The primary endpoint was the growth from birth to the term equivalent age and the secondary endpoint was the growth from birth to the age of 28 days.
This study deals with a topic of considerable interest and underlines the importance of a careful management of the premature infant during the critical phase of TF.
Despite the great relevance of this topic the study presents several issues that should be revised.
Comments:
Introduction:
Line 37-38. To argue that “the goal for nutritional care of VLBW infants is to achieve a growth rate similar to that of a fetus with similar gestational age in utero” is nowadays debated .
Line 43-44. Please add a reference.
Line 45. Please reverse “enteral to parenteral”
Materials and Methods:
Line 56. The authors included retrospectively infants born between 2005 and 2013. As the authors stated in the Nutrition practice section, since 2010 they changed their procedure increasing the daily protein intake. For this reason, it would be advisable for the authors to specify the nutritional intakes not only during the TF but also during the PN phase and EN phase.
Line 71. The authors used the Finnish growth charts. Were all the infants included Finnish? Why didn't they use curves with international impact?
Line 84. The use of hypothetical nutritional intakes is the main limitation of this study that aimed to explore the association between the length of the TF and macronutrient intake. In addition, consider similar macronutrient intakes for fresh own mother’s milk and pasteurized donor milk could be erroneous.
Line 94-95. I think that it is better to exclude the outliers rather than insert hypothetical data.
Line 104. Why the authors consider term equivalent age 35+ 2/7 to 43 6/7
Line 130. The author considered the TF as the period when enteral feed volumes were advancing between 20 and 120 ml/kg/d. It means that the PN was decreased as soon as the infants passed the MEF period whereas the fortification of human milk started when they achieve 100 ml/kg/day . I think that this is the cause of their nutritional deficit and poor growth. The authors should describe how the PN was decrease during the TF (both in terms of volume and above all in terms of intakes).
Results:
Line 168. It is curious to see that the included infants have such a low GA (mean 26.5 wks )and birth weight (mean 845 g) considering the inclusion criteria. How do the authors explain this result?
Line 184 and Figure 1. These results is very interesting. Could the authors also express these results as a daily intake (enteral and parenteral) in order to make these data more easily functional for the clinicians?
Figure 2. It is quite strange that the authors found differences in z-score between infants with intermediate TF and long TF, whereas no differences in z-score for HC and Weight in short TF vs the other two groups were found. I think that it may be due to the small sample size (Girl: 31 vs 38 vs 22 subjects; boys 23 vs 68 vs 40 subjects). Did the authors calculated a sample size powered for their outcomes?
Discussion:
In general, the discussion needs some modifications according to the previous comments.
Line 334-337.It is not so true that the feeding practice remained unchanged between 2005-2013. Indeed, the authors stated that the protein intakes for ELBW infants were increased since 2010 of around 1 g/kg/day. This could represent a relevant bias as the major part of infants included were ELBW.The authors should explain and comment this point.
Author Response
Reviewers' comments to Author and Author’s response
We are grateful for the constructive criticism of the reviewers. Please find below detailed information of all the changes made.
C1
Line 37-38. To argue that “the goal for nutritional care of VLBW infants is to achieve a growth rate similar to that of a fetus with similar gestational age in utero” is nowadays debated.
C2
Line 43-44. Please add a reference.
We thank for this comment and agree with the reviewer. The “traditional” goal for nutritional care of VLBW infant has recently been debated. We have now revised the introduction (lines 42-45) and added a reference as suggested: “A common goal for nutritional care of VLBW infants is to achieve growth similar to fetal growth, although this has recently been debated [3].”
C3
Line 45. Please reverse “enteral to parenteral”
We thank the reviewer for pointing out this mistake. This phrase has now been reversed (line 54).
C4
Line 56. The authors included retrospectively infants born between 2005 and 2013. As the authors stated in the Nutrition practice section, since 2010 they changed their procedure increasing the daily protein intake. For this reason, it would be advisable for the authors to specify the nutritional intakes not only during the TF but also during the PN phase and EN phase.
We thank the reviewer for this comment. According to this suggestion, we have now added a Supplemental Figure 1 and revised the results section (lines 216–219): “Energy and protein intake in the parenteral, transition, and enteral phase are shown in Supplemental Figure 1. Since the enteral feeding guidelines for ELBW infants were updated in 2010, the data are presented separately for infants born in 2005–2009 and 2010–2013. Infants born in these two eras had similar protein intake during the transition phase.“
C5
Line 71. The authors used the Finnish growth charts. Were all the infants included Finnish? Why didn't they use curves with international impact?
The Finnish growth charts are population-based multi-ethnic charts and they are based on the data of more than 500 000 newborns including infants of all ethnic backgrounds born in Finland, and included in the Finnish Medical Birth Register. They have been shown to be more accurate depicting newborn size and preterm growth in the Finnish population (regardless of ethnic background) than international charts (Sankilampi, U. et al. New population-based references for birth weight, length, and head circumference in singletons and twins from 23 to 43 gestation weeks. Ann. Med. 2013, 45, 446–454). The present study population included very preterm infants born under the gestational age of 32 weeks with birthweight under 1500 g and the proportion of infants with birthweight under 1000 g was 79%. For example, the international Intergrowth 21 growth charts for preterm infants is based on growth of only 201 preterm infants, of which only 12 were born before the gestational age of 32 weeks, and none before gestational age of 27 weeks (Villar J et al. Postnatal growth standards for preterm infants: the Preterm Postnatal Follow-up Study of the INTERGROWTH-21st Project. The Lancet Global Health. 2015;3(11):e681-91.). For comparison, the reference used in the present study is based on data of more than 3000 infants born under 32 gestation weeks. Thus, we feel that using the Finnish growth charts in this study is acceptable.
C6
Line 84. The use of hypothetical nutritional intakes is the main limitation of this study that aimed to explore the association between the length of the TF and macronutrient intake. In addition, consider similar macronutrient intakes for fresh own mother’s milk and pasteurized donor milk could be erroneous.
We agree that the macronutrient intake for fresh own mother’s milk and pasteurized donor milk is not comparable. Since the fresh own mother’s milk composition is also variable, it is challenging to control the nutrient composition of human milk especially in a retrospective study setting (Gidrewicz, D.A.; Fenton, T.R. A systematic review and meta-analysis of the nutrient content of preterm and term breast milk. BMC Pediatr. 2014, 14, 1–14). Unfortunately, during the study period the composition of human milk was not analyzed.
To clarify this, the methods section has now been revised (lines 105-109): “During the study period the composition of human milk was not analysed in our unit, therefore the content for human milk (mother’s own milk or donor milk) was based on the values reported in the literature: 67.7 kcal/100 mL, 1.4 g/100 mL, 3.2 g/100 mL, 8.4 g/100 mL for energy, protein, fat, and carbohydrates, respectively [14].”
We agree with the reviewer that this is a limitation of our study, and to stress this out we have revised the discussion section (line 400–402): “Also, a notable limiting factor is that the composition of the human milk used in the analyses was based on values reported in the literature and not on actual analyses. “
C7
Line 94-95. I think that it is better to exclude the outliers rather than insert hypothetical data.
We studied cumulative nutrient intake from birth to the age of 28 days in 248 very low birthweight infants (in total 6994 daily intakes). The nutritional data used were comprehensive and based on over 600 000 medication administration records in an electronic patient information system. There were only six days (0.09% of all days analyzed) with missing daily intake data and fourteen outliers (0.2% of all days analyzed) with nutritional intake 2–3 times higher or 8–10 times lower than the intake of the surrounding days of the respective infant. These gaps and outliers were mostly caused by missing recordings due to software updates of the electronic patient information system. The missing data and the outliers were processed similarly; interpolated using the surrounding measurements. Interpolation can be regarded as an adequate method as time series of the nutritional data were highly regular.
To clarify this, we have now revised the methods section to (line 11–118): “These defects and outliers were interpolated using the surrounding measurements of the respective infant. “
C8
Line 104. Why the authors consider term equivalent age 35+ 2/7 to 43 6/7
The growth parameters at term equivalent age were received from the Finnish Medical Birth Register. The register data are collected either until discharge from the hospital or gestational age of 42 + 0/7 weeks. Unfortunately, there were infants whose growth assessments were recorded after the age of 42+0/7 weeks. Also, there were infants who were discharged from the hospital before the age of 37+0/7 weeks. However, the median age for the term equivalent age growth assessment was 40+1/7 weeks (interquartile range 38+3/7–41+6/7).
To control for this variance of the postmenstrual age at term equivalent age growth assessments, the gestational age of this measurement was included in the analyses as a confounding factor (line 325, Table 3).
C9
Line 130. The author considered the TF as the period when enteral feed volumes were advancing between 20 and 120 ml/kg/d. It means that the PN was decreased as soon as the infants passed the MEF period whereas the fortification of human milk started when they achieve 100 ml/kg/day . I think that this is the cause of their nutritional deficit and poor growth. The authors should describe how the PN was decrease during the TF (both in terms of volume and above all in terms of intakes).
According to this comment and the previous comment (C4), we now revised the results section (lines 216–219) and added a Supplemental Figure 1 (lines 424–428). Figure S1: Energy and protein intake in the parental, transition and enteral phase in infants born in 2005–2009 and 2010–2013. The enteral nutrition guideline for ELBW infants (birthweight <1000 g) was updated in 2010. GAM method with defaulf parameters in geom_smooth() function of the ggplot2 package in R was used for smoothing.”
C10
Line 168. It is curious to see that the included infants have such a low GA (mean 26.5 wks) and birth weight (mean 845 g) considering the inclusion criteria. How do the authors explain this result?
In this study, we included infants who were hospitalized in our unit (a level 3 neonatal intensive care unit) for at least 28 days. In our unit, infants will be transferred to lower level units as soon as their condition allows that. Thus, only the most immature or otherwise sick infants will stay in our unit for a longer period of time. Thus, those infants staying for at least 28 d (and included in our study) will be the most vulnerable ones.
C11
Line 184 and Figure 1. These results is very interesting. Could the authors also express these results as a daily intake (enteral and parenteral) in order to make these data more easily functional for the clinicians?
We thank the reviewer for this comment. According to this and the previous comments (C4 and C9), we have now expressed the daily energy and protein intake (enteral, parenteral and total) in Supplemental Figure 1 (Figure S1).
C12
Figure 2. It is quite strange that the authors found differences in z-score between infants with intermediate TF and long TF, whereas no differences in z-score for HC and Weight in short TF vs the other two groups were found. I think that it may be due to the small sample size (Girl: 31 vs 38 vs 22 subjects; boys 23 vs 68 vs 40 subjects). Did the authors calculated a sample size powered for their outcomes?
Figure 2 shows the weight, length, and head circumference z-score change from birth to the age of one, two, three, and four weeks, and to term equivalent age in girls and boys in the three transition length-based groups. To compare the difference in z-score change between these three groups, linear mixed model were applied. In the model short transition phase group were set as a reference level. No comparisons between intermediate and long transition phase groups were made. To clarify the figure and the analyses made, we now revised the methods section (lines 183–185): In the model, infants were divided into three groups according to transition phase length quartiles, where the middle quartiles were combined into one group and the first quartile was set as reference level.
Also Figure 2 (line 294) was revised: Short transition length up to 7 days was set as reference level.
In this retrospective work were included all infants meeting the inclusion criteria (N=248). We agree that in the linear mixed model (Figure 2) the sample sizes of the subgroups were small. However, these results were in line with the results of the linear model (Line 326, Table 3) where associations between transition phase length and postnatal growth were studied in the whole population of 248 infants.
We did not perform power analysis. Compared to previously published studies of 156, 106 ans 95 VLBW or ELBW infants (reference number 18, 20, 28), the sample size of this study was notably larger.
C13
In general, the discussion needs some modifications according to the previous comments. Line 334-337.It is not so true that the feeding practice remained unchanged between 2005- 2013. Indeed, the authors stated that the protein intakes for ELBW infants were increased since 2010 of around 1 g/kg/day. This could represent a relevant bias as the major part of infants included were ELBW.The authors should explain and comment this point.
According to this and the previous comment (C4) we have revised the discussion section to (lines 398–400): “The local PN guidelines remained unchanged, however, the EN guideline was updated in 2010 and the target protein intake for ELBW infants increased. Yet, this did not affect the protein intake of these infants during the transition phase.”
Reviewer 3 Report
In this study authors aimed to study how the length of the transition phase from enteral to parenteral nutrition associates with macronutrient intake and postnatal growth in VLBW infants and found that the length of the transition phase associated negatively with postnatal macronutrient intake and growth in VLBW infants. For these infants, a rapid transition to full enteral feeding might be beneficial. Therefore, if enteral nutrition cannot be advanced, well-planned parenteral nutrition during the transition phase is necessary.
This is a very well-structured study whose results highlights the importance of neonatal nutrition. Increased nutritional intake during the transition phase from enteral to parenteral nutrition are associated with reduced extrauterine growth restriction and morbidity. The authors with their results reinforced the fact that there is room for improvement regarding nutritional intakes and growth in general and surrounding the transition phase from enteral to parenteral nutrition in particular.
Author Response
We thank the reviewer for this comment.
Round 2
Reviewer 2 Report
The authors adequately answered the comments and questions, therefore I believe that the manuscript is acceptable in its present form.